

# Association between laryngopharyngeal reflux and obstructive sleep apnea in adults

Susyana Tamin[1], Dumasari Siregar[1], Syahrial Marsinta Hutauruk[1], Ratna Dwi Restuti[1], Elvie Zulka Kautzia Rachmawati[1] and Saptawati Bardosono[2,†]

[1] Department of Otorhinolaryngology-Head and Neck Surgery, Faculty of Medicine, Universitas Indonesia, Jakarta, Indonesia
[2] Department of Nutrition Science, Faculty of Medicine, Universitas Indonesia, Jakarta, Indonesia
† Deceased.

## ABSTRACT

**Background**. Obstructive sleep apnea syndrome (OSAS) and laryngopharyngeal reflux (LPR) have been found to coexist in the population. OSAS and LPR also share obesity as an important risk factor. However, the relationship between LPR and OSAS remains unclear. This study aimed to correlate LPR as measured by the Reflux Symptom Index (RSI) and the Reflux Finding Score (RFS) with OSAS.

**Methods**. This cross-sectional study included sixty-four subjects who underwent anamnesis to complete the RSI and the Epworth Sleeping Scale (ESS). The subjects were then divided into the OSAS and non-OSAS groups based on the Apnea-Hypopnea Index (AHI) obtained through a polysomnography examination. Both groups underwent a flexible fiberoptic nasopharyngolaryngoscopy examination to determine the RFS. LPR was identified based on the RSI and RFS.

**Results**. The mean BMI of the OSAS group significantly was higher than the non-OSAS group ($p < 0.05$). Most of the subjects in the OSAS group exhibited mild-moderate OSAS (AHI 10–29), and severe OSAS occurred in only seven subjects. The mean RSI and RFS values in the OSAS group did not differ significantly from the non-OSAS group ($p = 0.34$ and $p = 0.36$, respectively). The proportion of LPR between the mild-moderate OSAS group, the severe OSAS group, and the non-OSAS group did not differ significantly ($p = 1.00$). RSI and RFS did not significantly correlate with AHI. Based on RSI, the proportion of LPR between the ESS (+) and ESS (-) groups did not significantly differ (adjusted $p = 0.062$). The proportion of LPR based on RFS was almost equal between the ESS (+) and ESS (-) groups ($p = 0.817$).

**Conclusions**. The BMI of the OSAS group was significantly higher than the non-OSAS group. There was no significant difference in RSI and RFS between the OSAS and non-OSAS groups. There was no significant correlation between RSI and AHI, or between RFS and AHI. There was no significant difference in the proportion of RSI between the ESS (+) and the ESS (-) groups.

Corresponding author
Susyana Tamin, usyana@yahoo.com

## INTRODUCTION

Obstructive sleep apnea syndrome (OSAS) is a sleep disorder characterized by repetitive episodes of temporary interruption of breathing during sleep, caused by recurrent upper respiratory airway obstruction due to inadequate dilator muscle tone. The cardinal symptoms include snoring, irritability, disruption of regular breathing patterns during sleep, morning headaches, and excessive sleepiness (*Mediano et al., 2019*; *Spicuzza, Caruso & Di Maria, 2015*). OSAS can lead to long-term health problems, such as metabolic disorders, cardiovascular disease, depression, and cognitive impairment. Moreover, OSAS is associated with lost productivity and sleep-related accidents (*Osman et al., 2018*). The diagnosis of OSAS is based on the parameters of the Apnea-Hypopnea Index (AHI), which is measured by polysomnography examination (PSG). This index measures the number of hypopnea or apnea events per hour. An AHI value $\geq 5$ is the cutoff used to diagnose OSAS. AHI values of 5–14 are classified as mild OSAS, 15–29 as moderate OSAS, and $\geq 30$ as severe OSAS. An AHI <5 is considered normal (*Arnold et al., 2017*; *Park, Ramar & Olson, 2011*).

Other than anatomical factors, the etiology of upper airway obstruction in patients with OSAS involves a failure of neuromuscular compensation. Using neurophysiological and histological approaches, several studies have shown sensory and motor function changes in patients with OSAS. In particular, the protective reflexes of the pharyngeal dilator muscle fail to respond to airway constriction during sleep. This may occur due to impaired receptor and pharyngeal sensitivity (afferent limb) or impaired motor nerves or muscles (efferent limb). This airway patency control impairment results in recurrent episodes of upper airway collapse during sleep (*Jobin et al., 2007*; *Saboisky et al., 2012*; *Saboisky et al., 2015*). Neuropathy and impaired pharyngeal sensitivity may be attributable to laryngopharyngeal reflux (LPR). Frequently, OSAS and LPR are found to coexist in the population (*Park, Ramar & Olson, 2011*). For example, a high incidence of LPR (45.2%) was found in patients with OSAS (*Magliulo et al., 2018*). The mechanism that explains this coexistence is the sensation of choking during sleep due to the microaspiration of stomach acid and the appearance of edema as a sign of inflammation leading to upper airway obstruction and neural damage. This neuropathy will eventually deteriorate the laryngeal dilator reflex and lead to apnea (*Park, Ramar & Olson, 2011*). In addition, both LPR and OSAS share obesity as an important risk factor. Obesity may induce reflux and apnea simultaneously by increasing intra-abdomen pressure and decreasing upper airway permeability (*Caparroz et al., 2019*).

To date, there has been no research correlating LPR with OSAS and comparing it with non-OSAS patients. Determining the correlation between LPR and OSAS is very important because the pathogenesis of OSAS remains unclear. Thus, the current study investigated the relationship between LPR events and OSAS. To this end, the Reflux Symptoms Index (RSI) and the Reflux Finding Score (RFS) were used measure LPR, and this was correlated with several OSAS parameters, such as excessive daytime sleepiness as assessed by the Epworth Sleeping Scale (ESS) and the AHI.

## MATERIALS & METHODS

### Participants and protocol

This cross-sectional study assessed the correlation between LPR and OSAS in taxi drivers. The study was conducted at the Snoring Clinic in the Endoscopic Broncho-Esophagology Division, Otorhinolaryngology–Head, and Neck Surgery Department, Faculty of Medicine Universitas Indonesia (FMUI)/Cipto Mangunkusumo Hospital (RSCM), Jakarta, Indonesia. The inclusion criteria included an age over 18 years complaints of snoring, and a willingness to participate in the study. The exclusion criteria included masses or tumors in the upper airway, including the nasal cavity, nasopharynx, oropharynx, hypopharynx, or pharynx. The Medical Research Ethics Committee in The Faculty of Medicine at Universitas Indonesia approved this study (No: 178/PT02.FK/ETIK/2010). The subjects provided verbal and written informed consent before participating.

Sixty-four subjects underwent anamnesis to complete the RSI and ESS. The subjects were then divided into OSAS and non-OSAS groups based on the AHI from the polysomnography. Both groups underwent flexible fiberoptic nasopharyngolaryngoscopy examinations to assess the RFS.

LPR was assessed using the RSI and RFS. For the RSI, the subjects were asked to report LPR symptoms, such as the presence of voice problems/hoarseness, throat clearing, difficulty in swallowing, postnasal drip, coughing after eating or lying down, choking, annoying cough, lump in the throat, and heartburn. Each RSI symptom was scored from 0 (no complaint) to 5 (severe complaint). The subjects were considered LPR based on RSI if the total score was more than 13. To evaluate the RFS, three independent ENT specialists or staff from the Endoscopic Broncho-Esophagology Division FMUI used flexible nasopharyngolaryngoscopy to assess subglottic edema, ventricular obliteration, erythema/hyperemia, diffuse laryngeal edema, vocal fold edema, tissue granulation, posterior commissure hypertrophy, and thick endolaryngeal mucus. The subjects were considered to have LPR if the RFS total score was more than 7.

OSAS was assessed using the AHI and ESS. To determine the AHI, the subjects underwent polysomnography, including a level 4 sleep test, which measures airflow and oximetry. The subjects were considered to have mild-moderate OSAS if the AHI score was 10–29, and severe OSAS if the AHI score was $\geq 30$. For the ESS, the subjects were asked to rate the chances of falling asleep in various situations, such as sitting/reading, watching television, sitting and talking to someone, sitting quietly after lunch, lying down to rest in the afternoon, in car while stopped for a few minutes in traffic, as a passenger in a car riding for an hour without a break, and sitting inactive in a public space. The score for each situation varied from 0 (would never doze) to 3 (high chance of dozing). The subjects were considered to have excessive daytime sleepiness if the ESS total score was more than 10.

### Statistical analysis

All statistical analyses were performed using the Statistical Package for Social Sciences (SPSS) software version 26 (IBM Corporation, Armonk, NY, USA). Variables such as age, Body Mass Index (BMI), and AHI score were tested for a normal distribution using the Kolmogorov–Smirnov test, and presented as mean values if they were distributed normally.

If they were not distributed normally, the data were presented as median values (min-max). If the data were distributed normally, unpaired t-tests were used to compare the mean differences in each variable between the OSAS and non-OSAS groups. If the data were not distributed normally, Mann–Whitney U tests were performed to compare the mean differences for each variable. The proportion differences in LPR between the OSAS and non-OSAS groups were tested using chi-squared or Fisher's exact tests with odds ratio (OR) values. The correlations between LPR indicators and OSAS indicators were examined using Pearson's correlation coefficients if the data were distributed normally, and Spearman's rank correlation coefficients if the data were not distributed normally. Multiple pairwise comparisons were performed using the Bonferroni correction to find the adjusted $p$-values. A $p$-value of $< 0.05$ was considered statistically significant.

## RESULTS

The ages of subjects were normally distributed, with mean ages of 43.38 years (SD 8.76) in the OSAS group and 42.75 years (SD 9.82) in the non-OSAS group. The mean ages between the two groups were not significantly different ($p = 0.159$, unpaired $t$-test). The mean BMI in the OSAS group was 26.23 (SD 3.71), and in the non-OSAS group it was 23.72 (SD 3.60). An unpaired $t$-test indicated that the mean BMI was significantly larger in the OSAS group compared to the non-OSAS group ($p = 0.005$; Table 1).

Based on the AHI score, most of the subjects in the OSAS group exhibited in mild-to-moderate OSAS (AHI 10–29), whereas severe OSAS occurred in only seven subjects. Based on the ESS results, more than half of the subjects, both in the OSAS and non-OSAS groups, were not suspected of having OSAS (ESS ≤10). The ESS scores of the OSAS group did not differ significantly from the non-OSAS group ($p = 0.564$ chi-squared test).

Table 2 shows that the mean RSI value in the OSAS group did not differ significantly from the non-OSAS group ($p = 0.34$, Mann–Whitney U test). The RFS values in the OSAS and non-OSAS groups were normally distributed. An unpaired $t$-test, indicated that the mean RFS in the OSAS group did not differ significantly from non-OSAS group ($p = 0.36$). A Fisher's exact test indicated that the proportion of LPR based on RFS in the OSAS group did not differ significantly from the non-OSAS group ($p = 1.00$).

Table 3 shows that although the LPR proportions were found to be highest in the non-OSAS group, the proportions of LPR between the mild-moderate OSAS group, severe OSAS group, and the non-OSAS group did not differ significantly ($p = 1.00$, Kolmogorov–Smirnov test).

Table 4 shows the correlations between the RSI and RFS scores and AHI in the OSAS group (AHI ≥ 10). Based on the Spearman's correlation, both the correlations between RSI and AHI and RFS with AHI were not significant ($p = 0.411$ and $p = 0.531$, respectively).

Based on the RSI, the proportion of LPR was significantly greater in the ESS (+) group than in the ESS (-) group ($p = 0.022$ chi-squared test; Table 5). The prevalence ratio value was 5 with a confidence of 95%. However, the adjusted $p$-value for the multiple comparisons using the Bonferroni correction was 0.062. This indicates that RSI is not significantly related to ESS. The proportion of LPR based on RFS was almost equal between

**Table 1  Age, Body Mass Index (BMI), Apnea-Hypopnea Index (AHI) scores, and Epworth Sleeping Scale (ESS) scores for each group.**

| Variable | OSAS ($n = 32$) | | Non-OSAS ($n = 32$) | | $p$-value | Adjusted $p$-value |
|---|---|---|---|---|---|---|
| | $n$ | % | $n$ | % | | |
| **1. Age (years)** | | | | | | |
| 18–45 | 17 | 53.1 | 20 | 62.5 | 0.159 | 0.713 |
| 46–55 | 14 | 43.8 | 8 | 25 | | |
| 56–65 | 1 | 3.1 | 4 | 12.5 | | |
| **2. BMI (kg/m$^2$)** | | | | | | |
| <18.5 | 1 | 3.1 | 2 | 6.3 | 0.005 | 0.001 |
| 18.5–23 | 4 | 12.5 | 12 | 37.5 | | |
| 23.1–27 | 10 | 31.3 | 14 | 43.8 | | |
| >27 | 17 | 53.1 | 4 | 12.5 | | |
| **3. AHI** | | | | | | |
| <10 | 0 | 0 | 32 | 100 | <0.001 | 0.000 |
| 10–29 | 25 | 78.1 | 0 | 0 | | |
| ≥30 | 7 | 21.9 | 0 | 0 | | |
| **4. ESS** | | | | | | |
| ESS >10 | 7 | 21.9 | 9 | 28.1 | 0.564 | 0.567 |
| ESS ≤ 10 | 25 | 78.1 | 23 | 71.9 | | |

Notes.
OSAS, Obstructive sleep apnea syndrome.

**Table 2  Reflux Symptoms Index (RSI), Reflux Finding Score (RFS), and proportion of laryngopharyngeal reflux (LPR) according to the RFS in the obstructive sleep apnea syndrome (OSAS) and non-OSAS groups.**

| | OSAS $n = 32$ | Non-OSAS $n = 32$ | $p$-value | Adjusted $p$-value |
|---|---|---|---|---|
| **1. RSI median** | 5.00 (0–24) | 4.50 (1–30) | 0.34 (Mann–Whitney U test) | 0.34 |
| **2. RFS means** | 11.13 (3.40) | 12.00 (4–14) | 0.36 (unpaired t-test) | 0.299 |
| **3. LPR proportion** | | | | |
| LPR | 28 (87.5%) | 29 (90.6%) | 1.00 (Fisher's exact test) | 0.891 |
| Non-LPR | 4 (12.5%) | 3 (9.4%) | | |

the ESS (+) and ESS (-) groups and did not differ significantly ($p = 0.817$, chi-squared test). This indicates that RFS is not significantly related to ESS.

## DISCUSSION

In this study, most of the subjects in the OSAS and non-OSAS groups were between the ages of 18–45 years. The proportions of ages in the OSAS group did not differ significantly from the non-OSAS group. As age increases, changes occur in the anatomical structure and function of the upper respiratory tract, including descent of the hyoid bone, a decrease in the size of the upper airway lumen, a reduction in the genioglossus muscular response, and a reduction in upper airway reflex sensitivity resulting in the collapse of the upper

**Table 3 Proportion of laryngopharyngeal reflux (LPR) based on the Reflux Finding Score in the obstructive sleep apnea syndrome (OSAS) and non-OSAS groups.**

| | OSAS ($n = 32$) | | Non-OSAS | $p$-value | Adjusted $p$-value |
|---|---|---|---|---|---|
| | Mild-moderate OSAS | Severe OSAS | | | |
| | ($n = 25$) | ($n = 7$) | ($n = 32$) | | |
| LPR | 22 (88%) | 6 (85.7%) | 29 (90.6%) | 1.00 (Kolmogorov–Smirnov test) | 0.891 |
| Non-LPR | 3 (12%) | 1 (14.3%) | 3 (9.4%) | | |

**Table 4 Correlations between the Reflux Symptoms Index (RSI), the Reflux Finding Score (RFS) and the Apnea-Hypopnea Index (AHI).**

| | AHI | |
|---|---|---|
| | r | p (Spearman) |
| RSI | 0.150 | 0.411 |
| RFS | −0.115 | 0.531 |

**Table 5 Proportion of laryngopharyngeal reflux (LPR) occurrence in Epworth Sleeping Scale (ESS) (+) and ESS (-) groups.**

| | ESS (+) | ESS (-) | OR | CI 95% | $p$-value | Adjusted $p$-value |
|---|---|---|---|---|---|---|
| | ($n = 6$) | ($n = 48$) | | | (chi-squared) | |
| According to RSI | | | | | | |
| LPR | 5 (31.3%) | 4 (8.3%) | 5 | 1.148–21.778 | 0.022 | 0.062 |
| Non-LPR | 11 (68.8%) | 44 (91.7%) | | | | |
| According to RFS | | | | | | |
| LPR | 14 (87.5%) | 43 (89.6%) | | | 0.817 | 1.000 |
| Non-LPR | 2 (12.5%) | 5 (10.4%) | | | | |

**Notes.**
CI, confidence interval; RFS, Reflux Finding Score; RSI, Reflux Symptoms Index.

airway (*McMillan & Morrell, 2016*). The higher prevalence of OSAS with age is also due to an increase in comorbid factors, such as congestive heart failure, diabetes, renal failure, and cognitive impairment (*e.g.*, dementia) (*Glasser et al., 2011*). The absence of age-related effects on OSAS in the current study may be due to the presence of other more potent risk factors, such as increased BMI.

In this study, obesity and being overweight were found to be higher in the OSAS group than in the non-OSAS group. In particular, there was a significant difference in BMI between the groups. Obesity is considered to be a major risk factor for the development of OSAS. The prevalence of OSAS in obese or severely obese patients is almost twice that of normal-weight patients. Furthermore, patients with mild OSAS who gain 10% of their initial weight have risk six times higher for worsening OSAS, while an equivalent weight loss can improve OSAS by more than 20% (*Sahasrabuddhe et al., 2017*). Obesity contributes

directly to the pathogenesis of OSAS through a reduction in resting lung volume, lung compliance, and functional residual capacity due to the deposition of fat in the thorax and abdomen. These factors increase the passive collapsibility of the airway by reducing caudal traction. In addition, an increase in neck circumference and the deposition of adipose tissue around the upper airway can lead to airway collapse in patients with obesity (*Ong et al., 2013*). LPR was found to be significantly correlated with OSAS in patients with obesity (*Rodrigues et al., 2014*). The OSAS patients with LPR have a higher BMI compared to LPR patients (*Magliulo et al., 2018*). It has also been reported that the prevalence of LPR signs and symptoms in OSAS patients is significantly higher in patients with obesity (*Xavier, Moraes & Eckley, 2013*).

The ESS (-) group were more common in both the OSAS and non-OSAS groups in this study. The proportion of ESS in the OSAS group did not differ significantly from ESS in the non-OSAS group. The ESS is a brief and practical questionnaire introduced in 1991 that has been validated as a method to measure daytime sleepiness and is used as a screening tool for OSAS (*Bonzelaar et al., 2017*). Although the ESS is short, easy to complete, and the most commonly used questionnaire to assess excessive daytime sleepiness, this scoring system is subjective and has limitations. For example, differences in the daily life activities mentioned in the ESS may affects the patients' answers. Thus, the ESS may not be meaningful way to assess excessive daytime sleepiness in all patients (*Thorarinsdottir et al., 2019*).

A correlation between OSAS and LPR has been reported in several studies. However, a meta-analysis by *Magliulo et al. (2018)* showed no correlation between the severity of AHI in OSAS patients and the presence of LPR. Similarly, a study by *Iannella et al. (2019)* investigated the presence of LPR in OSAS patients using the salivary concentration test (PEP-test), and did not show a correlation between the severity of apnea and the grade of salivary reflux. A cross-sectional study by *Xavier, Moraes & Eckley (2013)* showed that eighty-nine percent of patients with OSAS had signs and symptoms suggestive of LPR. *Eryılmaz et al. (2012)* also reported that LPR treatment in patients with OSAS reduces snoring, the Visual Analog Scale (VAS) and ESS, but does not improve polysomnographic parameters. Moreover, CPAP treatment in OSAS patients improves RSI and RFS but does not improve 24-h pH monitoring. The current study found that LPR was not significantly correlated with OSAS (Table 2) and did not correlate significantly with the degree of OSAS (Tables 3 and 4). These results differ from *Elhennawi, Ahmed & Abou-Halawa (2016)* study, which found that patients with severe OSAS have increased nocturnal LPR reflux episodes compared to patients with mild OSAS, and that the number of reflux episodes and total reflux duration during sleep is significantly correlated with the degree of OSAS. However, these studies have methodological differences. For example, our study performed an LPR comparison analysis between the OSAS and non-OSAS groups, whereas *Elhennawi, Ahmed & Abou-Halawa (2016)* study only analyzed LPR based on the severity of OSAS without comparisons to a non-OSAS group.

The degree of the relationship between LPR and OSAS is still an ongoing debate. Acid reflux is said to be associated with OSAS because reflux may increase arousal during sleep, resulting in excessive daytime sleepiness in patients with OSAS. Sensory mucosal laryngeal disorders are another mechanism that can explain the pathophysiological correlation

between LPR and OSAS. *Payne et al. (2016)* study, as cited by *Novakovic & MacKay (2015)*, found an endoscopic clinical feature of laryngeal inflammation associated with higher AHI scores, and a decrease in laryngeal mechanical sensitivity in patients with OSAS. In a previous study conducted by *Aviv et al. (2000)* which was also cited by *Novakovic & MacKay (2015)*, there was a decrease in laryngeal sensory function in patients with LPR detected using the same sensory testing method (*Novakovic & MacKay, 2015*). Therefore, there appears to be a good evidence supporting the pathophysiological correlations between the laryngeal inflammation associated with LPR, sensory function, and OSAS.

The current study found no correlation between RSI and AHI. *Altintaş et al. (2017)* performed a comparative analysis of depression between LPR and non-LPR groups in patients with OSAS and showed a positive correlation between RSI and AHI scores. However, Altintas et al.'s' study had some limitations, such as the lack of a validated RSI and relatively small sample size.

The present study also found no significant correlation between RSI and ESS after correcting the *p*-values for multiple comparisons. All of the *p*-values computed by the multiple pairwise comparison tests were higher than those for the chi-squared tests. This may happen because of a lack of statistical power, a weakly significant global effect, conservative multiple comparison tests, or a small number of subjects. In addition, according to the RFS parameters, this study did not find a significant correlation between RFS and ESS (Table 5).

The upper airway layer is not created to be exposed to acids and pepsin. On examination, the intranasal and/or intraoral tissue will appear reddish, and have an uneven surface with excessive mucus production. The swelling and damage of the tissues makes the airway more susceptible to collapse. Finally, an increase in upper airway obstruction mediated by LPR is strongly associated with insomnia, poor quality of sleep, and hypersomnolence (*Rouse, 2016*). Another mechanism is that nighttime reflux events promote arousal in the patient. A high arousal index is associated with poor sleep quality. Physiologically, hyperarousal can be associated with activation of the neuroendocrine system, including the autonomic nervous system and the hypothalamic-pituitary-adrenal axis. This arousal may lead to an increase in sympathetic activation, manifested by such things as increased heart rate or blood pressure (*Jung, Choung & Talley, 2010*).

## CONCLUSIONS

The BMI of the OSAS group was significantly higher than the non-OSAS group. There were no significant differences in mean RSI and RFS between the OSAS and non-OSAS groups. In the proportion of LPR based on RFS, no difference was found between the OSAS and the non-OSAS groups. There were also no significant correlations between RSI and AHI, or between RFS and AHI. In addition, there were no significant differences in the proportion of RSI between the ESS (+) group and the ESS (-) group. There was no significant difference in the proportion of RFS between the ESS (+) and the ESS (-) groups.

## ACKNOWLEDGEMENTS

We would like to thank the Department of Otorhinolaryngology-Head and Neck Surgery, Faculty of Medicine, University of Indonesia, and the Department of Nutrition, Faculty of Medicine, University of Indonesia, for supporting this study.

### Funding

The authors received no funding for this work.

### Competing Interests

The authors declare there are no competing interests.

### Author Contributions

- Susyana Tamin and Dumasari Siregar conceived and designed the experiments, performed the experiments, analyzed the data, prepared figures and/or tables, authored or reviewed drafts of the paper, and approved the final draft.
- Syahrial Marsinta Hutauruk and Saptawati Bardosono conceived and designed the experiments, analyzed the data, prepared figures and/or tables, authored or reviewed drafts of the paper, and approved the final draft.
- Ratna Dwi Restuti and Elvie Zulka Kautzia Rachmawati analyzed the data, prepared figures and/or tables, authored or reviewed drafts of the paper, and approved the final draft.

### Human Ethics

The following information was supplied relating to ethical approvals (i.e., approving body and any reference numbers):

The Committee of the Medical Research Ethics of The Faculty of Medicine, Universitas Indonesia, regarding the protection of human rights and welfare in medical research, approved this study. (Ethical number: 178/PT02.FK/ETIK/2010)

### Data Availability

The raw data are available in the Supplementary File.

### Supplemental Information

Supplemental information for this article can be found online at http://dx.doi.org/10.7717/peerj.13303#supplemental-information.

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
