# Peer review of "Association between laryngopharyngeal reflux and obstructive sleep apnea in adults"

_PeerJ, doi:10.7717/peerj.13303_

## Round 0.1 · original submission · Minor Revisions

· Academic Editor

Minor Revisions

The study provides a significant amount of data that could have important clinical implications in the association between laryngopharyngeal reflux and obstructive sleep apnea, but there are some criticisms raised by the reviewers that should be addressed, please pay attention to their comments.

·

Basic reporting

- This paper examines the association between laryngopharyngeal reflux and obstructive sleep apnea syndrome (OSAS) in adults.
- The paper reports the following major findings:
1. Negative findings in most of the comparisons between the OSAA (mild-moderate and severe) and non-OSAS groups including:
a). Mean Reflux Symptom Index (RSI),
b). Mean Reflux Finding Score(RFS),
c). The proportion of the laryngopharyngeal reflux (LPR),
d). RSI and RFS correlation with Apnea-Hypopnea Index (AHI).
e). The proportion of LPRs based on RFS between the Epworth Sleeping Scale (ESS) based groups.
2. Positive findings are:
a). The proportion of LPR based on RSI varies significantly between Epworth Sleeping Scale (ESS) groups.

The paper clearly conveys the literature, methods, and major findings. Although there are multiple spelling and grammatical errors, for example, in the abstract, "index" is spelled "indeks." Proofreading can easily correct these minor mistakes.
Overall, the paper is self-contained with relevant results to support the claims.

Experimental design

Strengths of the paper
1. The paper includes clearly describes the introduction, methods, and major findings.
2. The paper performs sound statistical analysis and reports the details enough to replicate the study.
3. Conclusions are supported by the evidence presented in the paper.

Weaknesses

1. The paper includes the following statement in the conclusion section, “In conclusion, LPR is more related to decreased quality of life of 262 patients with OSAS than to cause apnea or hypopnea.” However, there is no evidence presented in the paper to support this claim. The paper infers this indirectly from ESS, which reflects average sleep propensity in daily life (ASP). To sustain this claim, the paper should include the Quality of Life (QoL) measure or exclude/change this statement as appropriate.

Validity of the findings

As the paper compares multiple tests and results, I would suggest correcting the p-values for the multiple comparisons to prevent false-positive results. Although it won't affect negative findings, it will rule out false-positive findings especially with the significant findings reported in the proportion of LPR based on RSI varies significantly between Epworth Sleeping Scale (ESS) groups.

Additional comments

Although negative findings are the primary findings of this paper, the paper involves sound methodology and gives enough details to be replicable. I recommend acceptance after minor revisions.

·

Basic reporting

No comment.

Experimental design

No comment.

Validity of the findings

No comment.

Additional comments

This paper deals with an important as well as an interesting topic, and is well-designed and well-written. The methodology and results are statistically robust. Some suggestions for improvement are stated below.

1)The Methodology within the Abstract should be presented in a more meaningful way. The study design is not clear as provided in its present form.

2) The statistical methodology should be stated in detail under a separate subsection within the Materials and Methods section.

3)The term “samples” should be replaced by “subjects” or “patients”.

---

## Round 0.2 · accepted · Accept

· Academic Editor

Accept

All comments have been addressed.

·

Basic reporting

NA

Experimental design

NA

Validity of the findings

NA

Additional comments

My points raised during the previous review have been addressed by the authors. I have no further suggestions.

·

Basic reporting

No comment.

Experimental design

No comment.

Validity of the findings

No comment.

Additional comments

The revisions are satisfactory.